# BrGANs: Stabilizing GANs' Training Process with Brownian Motion Control

## Abstract

The training process of generative adversarial networks (GANs) is unstable and does not converge globally. In this paper, we propose a universal higher-order noise-based controller called Brownian Motion Controller (BMC) that is invariant to GANs' frameworks so that the training process of GANs is stabilized. Specifically, starting with the prototypical case of Dirac-GANs, we design a BMC and propose Dirac-BrGANs, which retrieve exactly the same but reachable optimal equilibrium regardless of GANs' framework. The optimal equilibrium of our Dirac-BrGANs' training system is globally unique and always exists. Furthermore, we give theoretical proof that the training process of Dirac-BrGANs achieves exponential stability almost surely for any arbitrary initial value and derive bounds for the rate of convergence. Then we extend our BMC to normal GANs and propose BrGANs. We provide numerical experiments showing that our BrGANs effectively stabilize GANs' training process and obtain state-of-the-art performance in terms of FID and inception score compared to other stabilizing methods.

## 1 Introduction

Generative Adversarial Networks (GANs) (Goodfellow et al., 2014) are popular deep learning based generative architecture. Given a multi-dimensional input dataset with unknown $P_{real}$, GANs can obtain an estimated $P_{model}$ and produce new entries that are as close to indistinguishable as possible from the input entries. For example, GANs can be used to generate images that look real to the human eyes (Wang et al., 2017). A GAN's architecture consists of two neural networks: a *generator* and a *discriminator*. The generator creates new elements that resemble the entries from the input dataset as closely as possible. The discriminator, on the other hand, aims to distinguish the (counterfeit) entries produced by the generator from the original members of the input dataset. The GAN's two networks can be modeled as a minimax problem; they compete against one another while striving to reach a *Nash-equilibrium*, an optimal solution where the generator can produce fake entries that are, from the point of view of the discriminator, in all respects indistinguishable from real ones.

Unfortunately, training GANs often suffers from instabilities. Previously, theoretical analysis has been conducted on GAN's training process. Fedus et al. (2018) argue that the traditional view of considering training GANs as minimizing the divergence of real distribution and model distribution is too restrictive and thus leads to instability. Arora et al. (2018) show that GANs training process does not lead generator to the desired distribution. Farnia & Ozdaglar (2020) suggest that current training methods of GANs do not always have Nash equilibrium, and Heusel et al. (2017a) is able to push GANs to converge to local Nash equilibrium using a two-time scale update rule (TTUR).

Many previous methods (Mescheder et al., 2018; Arjovsky & Bottou, 2017; Nagarajan & Kolter, 2017; Kodali et al., 2017) have investigated the causes of such instabilities and attempted to reduce them by introducing various changes to the GAN's architecture. However, as Mescheder et al. (2018) show in their study, where they analyze the convergence behavior of several GANs models, despite bringing significant improvements, GANs and its variations are still far from achieving stability in the general case.

To accomplish this goal, in our work, we design a Brownian Motion Control (BMC) using control theory and propose a universal model, BrGANs, to stabilize GANs' training process. We start with the prototypical Dirac-GAN (Mescheder et al., 2018) and analyze its system of training dynamics.

We then design a Brownian motion controller (BMC) on the training dynamic of Dirac-GAN in order to stabilize this system over time domain $t$. We generalize our BMC to normal GANs' setting and propose BrGANs.

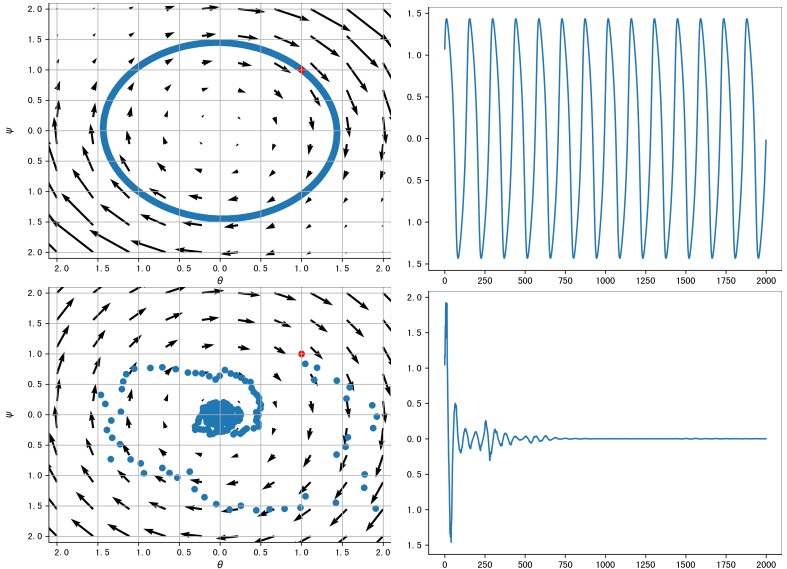

Figure 1: The gradient map and convergence behavior of Dirac-WGANs (first row) and Dirac-BrWGANs (second row), where the Nash equilibrium of both model should be at $(0,0)^T$.

## 1.1 SUMMARY OF OUR CONTRIBUTION

Compared with previous methods, we have the following contributions:

- We design Brownian Motion Controller (BMC), a universal higher order noise-based controller for GANs' training dynamics, which is compatible with all GANs' frameworks, and we give both theoretical and empirical analysis showing BMC effectively stabilizes GANs' training process.

- Under Dirac-GANs' setting, we propose Dirac-BrGANs and conduct theoretical stability analysis to derive bounds on the rate of convergence. Our proposed Dirac-BrGANs are able to converge globally with exponential stability.

- We extend BMC to normal GANs' settings and propose BrGANs. In experiments, our BrGANs converge faster and perform better in terms of inception scores and FID scores on CIFAR-10 and CelebA datasets than previous baselines in various GANs models.

## 1.2 RELATED WORK

To stabilize GANs training process, a lot of work has been done on modifying its training architecture. Karras et al. (2018) train the generator and the discriminator progressively to stabilize the training process. Wang et al. (2021) observe that during training, the discriminator converges faster and dominates the dynamics. They produce an attention map from the discriminator and use it to improve the spatial awareness of the generator. In this way, they push GANs' solution closer to the equilibrium.

On the other hand, many work stabilizes GANs' training process with modified objective functions. Kodali et al. (2017) add gradient penalty to their objective function to avoid local equilibrium with their model called DRAGAN. This method has fewer mode collapses and can be applied to a lot of GANs' frameworks. Other work, such as Generative Multi-Adversarial Network (GMAN) (Durugkar et al., 2017), packing GANs (PacGAN) (Lin et al., 2017) and energy-based GANs (Zhao et al., 2016), modifies the discriminator to achieve better stability.

Xu et al. (2019) formulate GANs as a system of differential equations and add closed-loop control (CLC) on a few variations of the GANs to enforce stability. However, the design of their controller depends on the objective function of the GANs models and does not work for all variations of the GANs models. Motivated by them, we analyze GANs' training process from control theory's perspective and design an invariant Brownian Motion Controller (BMC) to stabilize GANs training process. Compared with Xu et al. (2019), our proposed BrGANs converge faster, perform better, and do not rely on any specific GANs' architecture.

## 2 CONTROLLING DIRAC-GAN WITH BROWNIAN MOTION

In this section, we come up with the BMC, a higher order noise-based controller, as a universal control function that is invariant to objective functions of various GANs models. In addition, we prove that Dirac-GAN with BMC is exponentially stable and we derive bounds on its converge rate.

### 2.1 DYNAMIC SYSTEM OF DIRAC-GANS

In Dirac-GANs' settings, the distribution of generator G follows $p_G(x) = \delta(x - \theta)$ and the discriminator is linear $D_\phi(x) = \phi x$. The true data distribution is given by $p_D(x) = \delta(x - c)$ with a constant c. (Notice that for Dirac distribution, $\delta$ can be considered as an impulse on the origin such that $\delta(x) = 1$ at the origin and 0 otherwise.)

The objective functions of Dirac-GANs can be written as:

$$\begin{cases} \max_{\phi} L_{D_\phi}(\phi; \theta) = h_1(D_\phi(c)) + h_2(D_\phi(\theta)) \\ \max_{\theta} L_{G_\theta}(\theta; \phi) = h_3(D_\phi(\theta)), \end{cases} \tag{1}$$

where $h_1(\cdot)$ and $h_3(\cdot)$ are increasing functions and $h_2(\cdot)$ is a decreasing function around zero (Xu et al., 2019).

The training process of Dirac-GAN can be modelled as a system of differential equations. Following Mescheder et al. (2018) and Xu et al. (2019), the training dynamical system of Dirac-GAN is formulated as:

$$\begin{cases} \dfrac{\mathrm{d}\phi(t)}{\mathrm{d}t} = h_1'(\phi(t)c)c + h_2'(\phi(t)\theta(t))\theta(t), \\ \dfrac{\mathrm{d}\theta(t)}{\mathrm{d}t} = h_3'(\phi(t)\theta(t))\phi(t), \end{cases} \tag{2}$$

This system has a constant nontrivial solution $(\theta, \phi) = (c, 0)$.

Let $\tilde{\theta}(t) = \theta(t) - c$, and convert the original system (2) to:

$$\begin{cases} \dfrac{\mathrm{d}\phi(t)}{\mathrm{d}t} = h_1'(\phi(t)c)c + h_2'(\phi(t)(\tilde{\theta}(t) + c))(\tilde{\theta}(t) + c), \\ \dfrac{\mathrm{d}\tilde{\theta}(t)}{\mathrm{d}t} = h_3'(\phi(t)(\tilde{\theta}(t) + c))\phi(t). \end{cases} \tag{3}$$

At this time, the equilibrium of system (3) is $(0, 0)$.

Define $X(t) = (\phi(t), \tilde{\theta}(t))^\top$, $f(X(t)) = (h_1'(\phi(t)c)c + h_2'(\phi(t)(\tilde{\theta}(t) + c))\tilde{\theta}(t) + h_2'(\phi(t)(\tilde{\theta}(t) + c))c, h_3'(\phi(t)(\tilde{\theta}(t) + c))\phi(t))^\top$. Then system (3) can be rewritten as:

$$\mathrm{d}X(t) = f(X(t))\mathrm{d}t. \tag{4}$$

### 2.2 DESIGNING BROWNIAN MOTION CONTROLLER FOR DIRAC-GAN

Brownian motion is a natural phenomenon that captures the random displacements of particles in $d$-dimensional space. At each time step, the displacement $B_t$ is an independent, identical random variable ranging in $\mathbb{R}^d$. The distribution of $B_t$ is normally characterized by a multivariate Gaussian distribution.

Denote the position of a particle at initial time 0 as $X(0)$. Then at time $T$, this particle's position is given as

$$X(T) = X(0) + \int_0^T B_t \, dt. \tag{5}$$

In control theory, noise-based controllers like our Brownian motion controller (BMC) are a useful tool to stabilize dynamical systems and push the solution towards the optimal value over time domain $t$ (Mao et al., 2002). In this section, we design a BMC on Dirac-GAN's training dynamic to improve stability.

To stabilize system (4), we propose the following higher order noise-based controller:

$$u(t) = \varrho_1 X(t) \dot{B}_1(t) + \varrho_2 |X(t)|^\beta X(t) \dot{B}_2(t), \tag{6}$$

where $B_1(t)$ and $B_2(t)$ are independent one-dimensional Brownian motions, $\beta > 1$, $\varrho_1$ and $\varrho_2$ are non-negative constants. Incorporating BMC (6), the controlled system is given as

$$\mathrm{d}X(t) = f(X(t))\mathrm{d}t + u(t). \tag{7}$$

## 2.3 DIRAC-BRGAN WITH EXPONENTIAL STABILITY

In this section, we derive the existence of unique global solution and stability of system (7).

The equilibrium point of system (3) is

$$(\phi(t_e), \tilde{\theta}(t_e))^\top = (0, 0)^\top \tag{8}$$

Without the BMC, the training of a regular GAN or a WGAN (Arjovsky et al., 2017) is unstable and it oscillates around the equilibrium point $(0, 0)^\top$. Figure 1 illustrates the gradient maps of $\theta$ against $\phi$ and convergence behavior over time domain $t$. We can see that the gradients of both the generator and the discriminator are oscillating around the equilibrium point, but they never converge to it.

For the stability analysis, we impose the following assumption on the smoothness of functions $h_1$, $h_2$, $h_3$ in system (2).

**Assumption 1.** *There exist positive constants $\alpha_1$, $\alpha_2$, $\alpha_3$ such that for any $x, y \in \mathbb{R}^n$,*

$$|h_1'(x) - h_1'(y)| \leq \alpha_1 \|x - y\|, \quad |h_2'(x) - h_2'(y)| \leq \alpha_2 \|x - y\|,$$
$$|h_3'(x) - h_3'(y)| \leq \alpha_3 \|x - y\|.$$

In what follows, we first prove that the BMC from equation (6) yields a unique global solution in Theorem 1. That is, no matter which initial point $X(0)$ we are starting from, system (7) *a.s.* will have a unique solution $X(t)$ as $t$ goes to infinity. Then, in Theorem 2, we show that this unique global solution exponentially converges to the equilibrium point *a.s.* with bounds on the hyper-parameters $\varrho_1, \varrho_2$ and $\beta$ which in turn affect the rate of convergence. Combining Theorem 1 and Theorem 2, we claim that system (7) is stable and thus the Dirac-BrGAN is stable and converges to the optimal solution as required.

**Theorem 1.** *(Proof in Appendix A) Under Assumption 1, for any initial value $X(0) = \xi \in \mathbb{R}^2$, if $\varrho_2 \neq 0$ and $\beta > 1$, then there a.s. exists a unique global solution $X(t)$ to system (7) on $t \in [0, \infty)$.*

**Theorem 2.** *(Proof in Appendix B) Let Assumption 1 hold. Assume that $\varrho_2 \neq 0$ and $\beta > 1$. If*

$$\frac{\varrho_1^2}{2} - \varphi > 0,$$

*where $\varphi$ takes the value of*

$$\max_{x \geq 0} \left\{ -\frac{\varrho_2^2}{2} x^{2\beta} + (\alpha_2^2 + \frac{1}{2}\alpha_3^2)x^2 + [(1 + \frac{1}{2}\alpha_1^2)c^2 + 2c + \frac{1}{2}] \right\}, \tag{9}$$

*then for any $X(0) = \xi$ with sufficiently small constant $\epsilon \in (0, \varrho_1^2/2 - \varphi)$, the global solution $X(t)$ of system (7) has the property that*

$$\limsup_{t \to \infty} \frac{\log |X(t)|}{t} \leq -\left(\frac{\varrho_1^2}{2} - \varphi\right) + \epsilon, \quad a.s.$$

*that is, the solution of system (7) is a.s. exponentially stable.*

Here since $\frac{\varrho_1^2}{2} - \varphi > 0$ and $\epsilon$ is a sufficiently small constant, then when Eq. (9) is satisfied, we have

$$\limsup_{t \to \infty} \frac{\log |X(t)|}{t} \leq -\lambda, \quad a.s. \tag{10}$$

for some positive constant $\lambda$. Rearranging we get

$$\limsup_{t \to \infty} |X(t)| \leq e^{-\lambda t}, \tag{11}$$

which implies

$$\limsup_{t \to \infty} X(t) = (0, 0)^\top \tag{12}$$

as required. Notice that the rate of convergence depends only on constant $\lambda$, which in turn depends on $\varrho_1$ and $\varphi$. It means that the convergence rate is decided by the choice of hyper-parameters $\varrho_1$, $\varrho_2$, and $\beta$. In practice, we can tune these three variables as desired, as long as they satisfy the constraint from equation 9.

Notice that our Dirac-BrGANs works for any $h_1, h_2$ and $h_3$ as long as they satisfy the smoothness condition under assumption 1. In other words, we have proven that the Dirac-BrGANs are stable regardless of the GANs' architecture, and we have given theoretical bounds on the rate of convergence. In Figure 1, we present visual proof that the Dirac-BrGANs are stable and converge to the optimal equilibrum as required.

## 3 GENERALIZATION OF BMC TO GANS

In section 2 we designed a universal BMC for Dirac-GANs and proved that the Dirac-BrGANs are globally exponentially stable. In this section, we are going to generalize the BMC for normal GANs (i.e., GANs other than the Dirac-GAN). We consider any GANs where the Generator ($G$) and the Discriminator ($D$) are neural networks in their respective function spaces.

### 3.1 MODELLING DYNAMICS OF GANS

Analogously to the Dirac-GAN, the training dynamics of normal GANs can be formulated as a system of differential equations. Instead of $\theta$ and $\phi$, we directly start with $G(z, t)$ and $D(x, t)$ to represent, respectively, the generator and the discriminator. The objective functions of GANs can be written as:

$$\begin{cases} \max_D L_D(D; G) = \mathbb{E}_{p(x)}[h_1(D(x))] + \mathbb{E}_{p(g)}[h_2(D(x))] \\ \max_G L_G(G; D) = \mathbb{E}_{p_z(z)}[h_3(D(G(z)))]. \end{cases} \tag{13}$$

Following Xu et al. (2019)'s notation, the training dynamic for the generator and discriminator over the time domain $t$ can be transformed to:

$$\begin{cases} \dfrac{\mathrm{d}D(x, t)}{\mathrm{d}t} = p(x) \dfrac{\mathrm{d}h_1(D(x))}{\mathrm{d}D(x, t)} + p_G(x) \dfrac{\mathrm{d}h_2(D(x))}{\mathrm{d}D(x)}, \forall x \\ \dfrac{\mathrm{d}G(z, t)}{\mathrm{d}t} = p_z(z) \dfrac{\mathrm{d}h_3(D(G(z)))}{\mathrm{d}D(G(z))} \dfrac{\mathrm{d}D(G(z))}{\mathrm{d}G(z)}, \forall z \end{cases} \tag{14}$$

Now we define $X(t) = (D(x, t), G(z, t))^\top$ and

$$f(X(t)) = \begin{pmatrix} p(x) \frac{\mathrm{d}h_1(D(x))}{\mathrm{d}D(x,t)} + p_G(x) \frac{\mathrm{d}h_2(D(x))}{\mathrm{d}D(x)} \\ p_z(z) \frac{\mathrm{d}h_3(D(G(z)))}{\mathrm{d}D(G(z))} \frac{\mathrm{d}D(G(z))}{\mathrm{d}G(z)} \end{pmatrix}. \tag{15}$$

Now we convert system (14) to

$$\mathrm{d}X(t) = f(X(t))\mathrm{d}t. \tag{16}$$

## 3.2 BrGAN: Stabilized GANs with BMC

The optimal solution of a normal GAN is achieved when $D(x) = 0$ for the discriminator and $p_G(x) = p(x)$ for the generator. In control theory, when we design a controller for a dynamical system, we need to know the optimal solution so that we can use the controller to push this dynamical system to its equilibrium point without changing it. With normal GANs, we only have information on the optimal solution for the discriminator, so we are going to impose the BMC only on the discriminator this time.

Notice that in Dirac-GAN, we impose the BMC $u(t) = \varrho_1 X(t)\dot{B}_1(t) + \varrho_2|X(t)|^\beta X(t)\dot{B}_2(t)$ on $X(t)$, which is for both the generator and discriminator. Since now we are going to impose BMC only on the discriminator, without losing information from the generator, here we slightly modify Eq. (6) so that

$$u_D(t) = \varrho_1 D(x)\dot{B}_1(t) + \varrho_2(D^2(x) + D^2(G(z)))D(x)\dot{B}_2(t), \qquad (17)$$

where $B_1(t)$ and $B_2(t)$ are independent one-dimensional Brownian motions, $\varrho_1$ and $\varrho_2$ are non-negative constants.

Since we are going to implement BrGANs through gradient descent, our BMC can be reflected on the discriminator's objective function with the derivative being Eq. (17). We thus take integration of Eq. (17) and modify the objective function of the discriminator in (13) to:

$$\max_D L_D'(D;G) = L_D(D;G) + \frac{1}{2}\varrho_1 D^2(x)\dot{B}_1(t) + [\frac{1}{4}\varrho_2 D^4(x) + \frac{1}{2}\varrho_2 D^2(G(z))D^2(x)]\dot{B}_2(t). \qquad (18)$$

We implement our designed objective functions in section 4. Our numerical experiments show that BrGANs successfully stabilize GANs models and are able to generate images with promising quality.

## 4 Evaluation

In this section, we show the effectiveness of BMC by providing both quantitive and qualititive results.

### 4.1 Exeperimental setting

**Dataset:** We evaluate our proposed BrGANs on well-established CIFAR10 (Krizhevsky et al., 2009) and CelebA datasets (Liu et al., 2015). The CIFAR-10 dataset consists of 60000 32×32 color images in 10 classes, with 6000 images per class. There are 50000 training images and 10000 test images. This dataset can be used for both conditional image generation and unconditional image generation. In order to compare our training method fairly with the solutions from the related works, we use a batch size of 64, the same generator, and discriminator architecture under the same codebase. CelebA contains 202,599 face images of size 64 × 64, which has diverse facial features.

**Implementation details:** Both our generator and discriminator are composed of convolutional layers, batch normalization, and activation layers. The generator uses four layers of transposed convolutional layers to convert $1 \times 100$ latent vector to a $3 \times 32 \times 32$ image. Batch normalization and ReLU activation are followed by each layer. For the discriminator, first, we use three layers of convolutional layer to obtain $1024 \times 4 \times 4$ image and then feed this image to an MLP to get a single value.

Each of our models is trained on an Nvidia 2080TI GPU. The batch size is 64, while the generator is trained for 50000 iterations, and the discriminator is trained for 250000 iterations. We use Adam with 1e-4 learning rate as an optimizer to train our model.

**Evaluation Metric:** For the Dirac-GAN problem, the optimal solution is known to us so we can measure the convergence speed and draw the gradient map for different training algorithms. For CIFAR 10, we use the FID score (Heusel et al., 2017b) and the inception score(Barratt & Sharma, 2018) to measure the quality of the generated images. We also compare the FID and inception scores across different timestamps to show the convergence speed.

## 4.2 CONVERGENCE OF DIRAC-BRGAN

The gradient map and the convergence curve are presented in Figure 1. These results show that our Dirac-BrGANs have better convergence patterns and speed than Dirac-GANs. Without adding BMC to the training objective, Dirac-GANs cannot reach equilibrium. The parameters of the generator and the discriminator are oscillating in a circle as shown in figure 1. However, the parameters of the Dirac-BrGANs only oscillate in the first 500 iterations and soon converge in 800 iterations.

We also study different combinations of $\varrho_1$ and $\varrho_2$ under $\beta = 1$ and $\beta = 2$. As shown in table 1 and table 2, Dirac-BrGANs converge better when we set $\varrho_1 = 0.1$ and $\varrho_2 = 0.01$. Generally, a larger $\varrho$ will lead to a faster convergence rate but when $\varrho$ is large enough, the effect of increasing $\varrho$ will become saturated. On the other hand, when $\varrho$ is too small, Dirac-BrGANs will take more than 100000 iterations to converge.

Table 1: Convergence iters for $\beta = 2$ under Dirac-BrGANs

| $\beta = 2$ | $\varrho_2 = 0.0001$ | $\varrho_2 = 0.001$ | $\varrho_2 = 0.01$ |
|---|---|---|---|
| $\varrho_1 = 0.1$ | 1500 | 750 | 700 |
| $\varrho_1 = 0.01$ | >100000 | 9000 | 8500 |
| $\varrho_1 = 0.001$ | >100000 | >100000 | >100000 |

Table 2: Converge iters for $\beta = 1$ under Dirac-BrGANs

| $\beta = 1$ | $\varrho_2 = 0.0001$ | $\varrho_2 = 0.001$ | $\varrho_2 = 0.01$ |
|---|---|---|---|
| $\varrho_1 = 0.1$ | 600 | 400 | 400 |
| $\varrho_1 = 0.01$ | 25000 | 15000 | 10000 |
| $\varrho_1 = 0.001$ | >100000 | >100000 | 40000 |

## 4.3 STABILIZED BRGANS: CONVERGE FASTER AND PERFORM BETTER

We demonstrate that BrGANs converge fast and generate higher fidelity image than Wasserstein GANs (WGAN) (Arjovsky et al., 2017), WGAN with weight clipping (WGAN-CP), WGAN with gredient penlty (WGAN-GP) (Gulrajani et al., 2017), GAN with closed loop control (WGAN-CLC) (Xu et al., 2019) and their combinations. From the results in Table 3, we can observe that our proposed WGAN-BR-GP achieves 22.10 FID and 5.42 inception score on CIFAR10 dataset, which is the best result among all other GANs. Specifically, WGAN-BR out performs WGAN, WGAN-BR-CP outperforms WGAN-CP and WGAN-CLC-CP, and WGAN-BR-GP outperforms WGAN-GP and WGAN-CLC-GP. From Table 4, we can observe similar trends

Table 3: Results on CIFAR10.

| Method | FID | Inception |
|---|---|---|
| WGAN | 94.77 | 3.29 |
| WGAN-CLC | 52.39 | 4.25 |
| **WGAN-BR** | 36.50 | 4.80 |
| WGAN-CP | 37.81 | 4.69 |
| WGAN-CLC-CP | 35.59 | 4.64 |
| **WGAN-BR-CP** | 34.18 | 5.03 |
| WGAN-GP | 30.81 | 5.03 |
| WGAN-CLC-GP | 53.47 | 4.24 |
| **WGAN-BR-GP** | **22.10** | **5.42** |

Table 4: Results on CelebA.

| Method | FID | Inception |
|---|---|---|
| WGAN | 366.12 | 2.06 |
| WGAN-CLC | 14.87 | 3.28 |
| **WGAN-BR** | 23.91 | 3.34 |
| WGAN-CP | 13.60 | 3.14 |
| WGAN-CLC-CP | 13.80 | 3.12 |
| **WGAN-BR-CP** | 14.74 | 3.19 |
| WGAN-GP | 8.30 | 3.09 |
| WGAN-CLC-GP | 61.94 | **3.42** |
| **WGAN-BR-GP** | **6.20** | 2.91 |

Convergence iterations of GANs is presented in Fig. 2 and Fig. 3, measured by FID score and inception score, respectively. It is readily observed that our proposed BrGANs achieves better FID and inception scores given the same training iteration. Xu et al. (2019) add a L2 regularization

(CLC) on the objective function on discriminator. In our BrGANs, we incorporate both information from the generator and discriminator to our controller, so that the discriminator and generator to make sure the discriminator does not dominate the training process. Our BrGANs also compute faster than Xu et al. (2019) since we do not need to keep a buffer and update accordingly during training process.

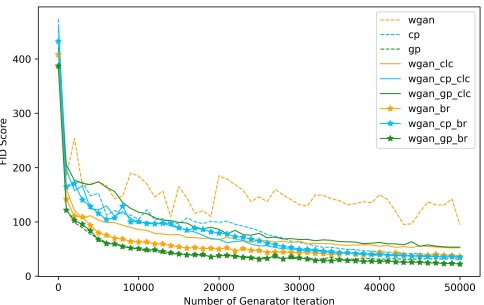 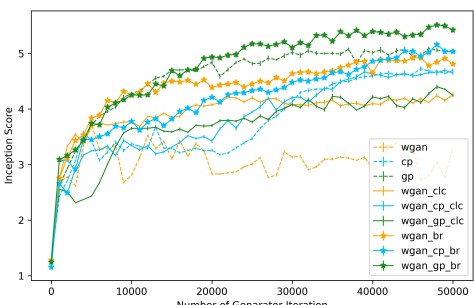

Figure 2: FID score on CIFAR10.      Figure 3: Inception score on CIFAR10.

## 4.4 QUALITATIVE RESULTS

We provide qualitative results on CIFAR and CelebA datasets. In Fig. 5 and Fig. 4, these images are generated by WGAN, WGAN-GP, WGAN-GP-BR, and WGAN-GP-CLC, from top left to bottom right respectively. It can be observed that our WGAN-GP-BR can generate images with higher fidelity and more reasonable in visual perception.

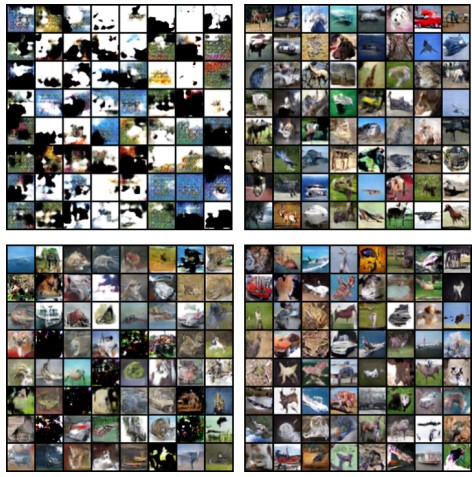 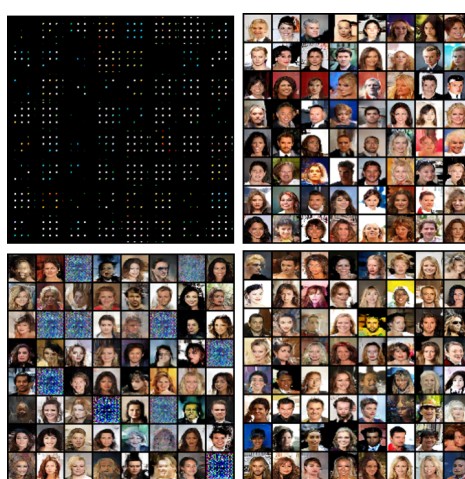

Figure 4: CIFAR           Figure 5: CelebA

## 5 CONCLUSION AND DISCUSSION

In this paper, we revisit GANs' instability problem from the perspective of control theory. Our work novelty incorporates a higher order non-linear controller and modify the objective function of the discriminator to stabilize GANs models. We innovatively design a universal noise-based control method called Brownian Motion Control (BMC) and propose BrGANs to achieve exponential stability. Notably, our BMC is compatible with all GANs variations. Experimental results demonstrate that our BrGANs converge faster and and perform better in terms of FID and inception scores on CIFAR-10 and CelebA.

In our paper, theoretical analysis has been done under Dirac-GANs' setting and we are able to stabilize both generator and discriminator simultaneously. However, under normal GANs' settings, we only design BMC for the discriminator and stabilize the discriminator, then force the stability of the generator. Additionally, our BMC is derived under continuous setting, but GANs' training process is considered as discrete time steps. To resolve these two problems, further work can be done on estimating the generator's equilibrium at each time step and imposing a controller on both generators and discriminators simultaneously.

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

## APPENDIX A: PROOF OF THEOREM 1

Under Assumption 1, for any initial value $X(0) = \xi \in \mathbb{R}^2$, if $\varrho_2 \neq 0$ and $\beta > 1$, then there a.s. exists a unique global solution $X(t)$ to system (7) on $t \in [0, \infty)$.

*Proof.* Under Assumption 1, then, we can calculate that

$$
\begin{aligned}
&X^{\mathrm{T}}(t)f(X(t)) \\
=&\phi(t)h_1^{'}(\phi(t)c)c + \phi(t)h_2^{'}(\phi(t)(\tilde{\theta}(t)+c))\tilde{\theta}(t) + \\
&\phi(t)h_2^{'}(\phi(t)(\tilde{\theta}(t)+c))c + \\
&\tilde{\theta}(t)h_3^{'}(\phi(t)(\tilde{\theta}(t)+c))\phi(t) \\
\leq&[(1+\frac{1}{2}\alpha_1^2)c^2 + 2c + \frac{1}{2}]|X|^2 + (\alpha_2^2 + \frac{1}{2}\alpha_3^2)|X|^4.
\end{aligned}
$$

For any bounded initial value $X(0) \in \mathbb{R}^n$, there exists a unique maximal local strong solution $X(t)$ of system (7) on $t \in [0, \tau_e)$, where $\tau_e$ is the explosion time. To show that the solution is actually global, we only need to prove that $\tau_e = \infty$ a.s. Let $k_0$ be a sufficiently large positive number such that $|X(0)| < k_0$. For each integer $k \geq k_0$, define the stopping time

$$
\tau_k = \inf\{t \in [0, \tau_e) : |X(t)| \geq k\}
$$

with the traditional setting $\inf \emptyset = \infty$, where $\emptyset$ denotes the empty set. Clearly, $\tau_k$ is increasing as $k \to \infty$ and $\tau_k \to \tau_\infty \leq \tau_e$ a.s. If we can show that $\tau_\infty = \infty$, then $\tau_e = \infty$ a.s., which implies the desired result. This is also equivalent to prove that, for any $t > 0$, $\mathbb{P}(\tau_k \leq t) \to 0$ as $k \to \infty$. To prove this statement, for any $p \in (0, 1)$, define a $C^2$-function

$$
V(x) = |X(t)|^p.
$$

One can obtain that $X(t) \neq 0$ for all $0 \leq t \leq \tau_e$ a.s. Thus, one can apply the Itô formula to show that for any $t \in [0, \tau_e)$,

$$
\begin{aligned}
\mathrm{d}V(X(t)) =&LV(X(t))\mathrm{d}t + p\varrho_1|X(t)|^p\mathrm{d}B_1(t) \\
&+ p\varrho_2|X(t)|^{\beta+p}\mathrm{d}B_2(t),
\end{aligned}
$$

where $LV$ is defined as

$$
\begin{aligned}
LV(X) =&p|X|^{p-2}X^{\mathrm{T}}f(X(t)) + \frac{p(p-1)\varrho_1^2}{2}|X|^p \\
&+ \frac{p(p-1)\varrho_2^2}{2}|X|^{2\beta+p}
\end{aligned}
$$

By Assumption 1, we therefore have

$$LV(X) \leq \frac{p(p-1)\varrho_2^2}{2}|X|^{2\beta+p} + ((1+\frac{1}{2}\alpha_1^2)c^2 + 2c + \frac{1}{2})$$
$$p|X|^{\alpha+p} + p\left(\frac{(p-1)\varrho_1^2}{2} + (\alpha_2^2 + \frac{1}{2}\alpha_3^2)\right)|X|^p.$$

Noting that $p \in (0,1)$ and $\beta > 1$ and $\varrho_2 \neq 0$, by the boundedness of polynomial functions, there exists a positive constant $\bar{H}$ such that $LV(x) \leq \bar{H}$. We therefore have

$$\mathbb{E}V(X(t \wedge \tau_k)) \leq \mathbb{E}|\xi|^p + \mathbb{E}\int_0^{t\wedge\tau_k} LV(X(s))\mathrm{d}s$$
$$\leq \mathbb{E}|\xi|^p + \bar{H}t$$
$$=: \bar{H}_t,$$

where $\bar{H}_t$ is independent of $k$. By the definition of $\tau_k$, $|X(\tau_k)| = k$, so

$$\mathbb{P}(\tau_k \leq t)k^p \leq \mathbb{P}(\tau_k \leq t)V(X(\tau_k))]$$
$$\leq \mathbb{E}[l_{\{\tau_k \leq t\}}V(X(t \wedge \tau_k))]$$
$$\leq \mathbb{E}V(X(t \wedge \tau_k))$$
$$\leq \bar{H}_t,$$

which implies that

$$\limsup_{k\to\infty} \mathbb{P}(\tau_k \leq t) \leq \lim_{k\to\infty} \frac{\bar{H}_t}{k^p} = 0,$$

as required. □

## APPENDIX B: PROOF OF THEOREM 2

Let Assumption 1 hold. Assume that $\varrho_2 \neq 0$ and $\beta > 1$. If

$$\frac{\varrho_1^2}{2} - \varphi > 0,$$

where

$$\varphi = \max_{x\geq0}\left\{-\frac{\varrho_2^2}{2}x^{2\beta} + (\alpha_2^2 + \frac{1}{2}\alpha_3^2)x^2 + [(1+\frac{1}{2}\alpha_1^2)c^2 + 2c + \frac{1}{2}]\right\}, \tag{19}$$

then for any $X(0) = \xi$ with sufficiently small constant $\epsilon \in (0, \varrho_1^2/2 - \varphi)$, the global solution $X(t)$ of system (7) has the property that

$$\limsup_{t\to\infty} \frac{\log|X(t)|}{t} \leq -\left(\frac{\varrho_1^2}{2} - \varphi\right) + \epsilon, \quad a.s.$$

that is, the solution of system (7) is a.s. exponentially stable.

*Proof.* Applying Itô formula to $\log|X(t)|$ yields

$$\log|X(t)| = \log|X(0)| + \int_0^t\left[|X(t)|^{-2}X^\mathrm{T}(s)f(X(s))\right.$$
$$\left. -\frac{\varrho_2^2}{2}|X(s)|^{2\beta} - \frac{\varrho_1^2}{2}\right]\mathrm{d}s + \int_0^t \varrho_1\mathrm{d}B_1(t)$$
$$+ \varrho_2\int_0^t |X(s)|^\beta\mathrm{d}B_2(s).$$

Letting $M(t) = \varrho_2 \int_0^t |X(s)|^\beta \mathrm{d}B_2(s)$, clearly $M(t)$ is a continuous local martingale with the quadratic variation

$$< M(t), M(t) >= \varrho_2^2 \int_0^t |X(s)|^{2\beta} \mathrm{d}s.$$

For any $\varepsilon \in (0, 1)$, choose $\theta > 0$ such that $\theta \varepsilon > 1$. Then for each integer $m > 0$, the exponential martingale inequality gives

$$\mathbb{P}\left\{ \sup_{1 \leq t \leq m} \left[ M(t) - \frac{\varepsilon \varrho_2^2}{2} \int_0^t |X(s)|^{2\beta} \mathrm{d}s \right] \geq \theta \varepsilon \log m \right\} \leq \frac{1}{m^{\theta \varepsilon}}.$$

Since $\sum_{m=1}^\infty m^{-\theta \varepsilon} < \infty$, by the well-known Borel-Cantelli lemma, there exists an $\bar{\Omega}_0 \subseteq \Omega$ with $\mathbb{P}(\bar{\Omega}_0) = 1$ such that for any $\omega \in \bar{\Omega}_0$, there exists an integer $\bar{m}(\omega)$, when $m > \bar{m}(\omega)$, and $m - 1 \leq t \leq m$,

$$M(t) \leq \frac{\varepsilon \varrho_2^2}{2} \int_0^t |X(s)|^{2\beta} \mathrm{d}s + \theta \varepsilon \log(t + 1).$$

This, together with Assumption 1, yields

$$\begin{aligned}
\log |X(t)| \leq \log |\xi| + \int_0^t &\left[ -\frac{\varrho_2^2 (1 - \varepsilon)}{2} |X(s)|^{2\beta} \right. \\
&+ (1 + (\frac{1}{2}\alpha_1^2)c^2 + 2c + \frac{1}{2})|X(s)|^\alpha \\
&\left. + (\alpha_2^2 + \frac{1}{2}\alpha_3^2) - \frac{\varrho_1^2}{2} \right] \mathrm{d}s \\
&+ \int_0^t \varrho_1 \mathrm{d}B_1(t) + \theta \varepsilon \log(t + 1).
\end{aligned}$$

Letting $\epsilon$ be sufficiently small, by the definition of $\varphi$ in (9), for sufficiently small $\epsilon \in (0, \varrho_1^2/2 - \varphi)$, we have

$$\begin{aligned}
\log |X(t)| \leq \log |\xi| &- \left[ (\frac{\varrho_1^2}{2} - \varphi) - \epsilon \right] t + \int_0^t \varrho_1 \mathrm{d}B_1(t) \\
&+ \theta \varepsilon \log(t + 1).
\end{aligned}$$

Applying the strong law of large number, we therefore have

$$\limsup_{t \to \infty} \frac{\log |X(t)|}{t} \leq -(\frac{\varrho_1^2}{2} - \varphi) + \epsilon \quad a.s.$$

$\square$

