# OpenReview forum: "BrGANs: Stabilizing GANs' Training Process with Brownian Motion Control"
_ICLR.cc/2023/Conference — Submitted to ICLR 2023_

### Official Review · Reviewer_Vy2E · 2022-10-24

**Confidence:** 3
**Clarity, Quality, Novelty And Reproducibility:** Everything except evaluation quality …
**Correctness:** 3
**Technical Novelty And Significance:** 3
**Empirical Novelty And Significance:** 3
**Recommendation:** 5

**Strength And Weaknesses:**

Strengths

If the main weaknesses below are fixed and the papers empirical improvements persists, the paper becomes an incremental improvement over CLC gan which will spread awareness of BMC theory in the community, which I think is absolutely enough. I'd also be interested in possible links to diffusion processes.

Weaknesses

1. the FID of WGAN-CLC-GP is much higher than in the Xu et al paper, but the IS is much closer (still higher). To me this indicates FID was calculated with a different batch size than in the Xu et al paper *but* also that the improvement might be due to architecture/training differences. An ablation study might alleviate this...
2. ....As would reporting multiple seeds, reporting mean and std and performing a statistical significance test
3. the novelty relies on adding gaussian noise to the control process + an analysis, which I think is just barely enough to warrant a paper, but is sabotaged by limited evaluation quality. If the evaluation is fixed and benefits persist, this weakness becomes irrelevant.


**Summary Of The Paper:**

The paper extends the CLC-GAN framework of Xu et al 2019 https://proceedings.mlr.press/v119/xu20d.html with a gaussian noise regularisation derived from Brownian Model Control theory, showing that this additionally improves over the CLC controller in terms of FID on CIAR10 and celeba

**Summary Of The Review:**

As noted, if the empirical gains survive an improved evaluation, I think this paper is a fine incremental improvement over previous work, using not-so widely known BMC theory

---

> ### Author Response · Authors · 2022-11-17
> **Thank your for your supportive comments**
>
> We appreciate the acknowledgement of the innovation of our method. We address the detailed response below. We hope you find your response satisfactory and raise your score accordingly.
>
> __Q1 FID and IS scores different from CLC__
> See general feedback A1.
>
> __Q2 Multiple seed and statistical significance test__
> Each of our experiments is performed 3 times. We report the mean and standard deviation of the result (see general feedback A1). In our original experiment, the scatter graph includes an error bar for each point. To clarify, we will include mean and std values in our table for our supplementary experiment.

---

> > ### Comment · Reviewer_Vy2E · 2022-12-03
> > **Thank your for the response**
> >
> > Dear authors,
> >
> > I thank you for responding to me and the other authors. I'm happy to see that you could align the scores with the CLC setting and that it seems the improvement seems to persist, albeit apparently on a much smaller scale? I think, as has been raised by other reviewers, that this will require to add more investigations and/or explanation of *why* it helps training (is it a slightly randomized step size which prevents cycling? ) and what it adds over CLC alone to make the paper as strong as it could be.

---

### Official Review · Reviewer_YEms · 2022-10-24

**Confidence:** 4
**Correctness:** 3
**Technical Novelty And Significance:** 3
**Empirical Novelty And Significance:** 2
**Recommendation:** 5

**Clarity, Quality, Novelty And Reproducibility:**

I think more intuition could be given to eq (6). Also what does it mean dot B_1, is it dB_1? What is p(g) in eq (13), does that include the GP penalty of WGAN-GP? In section 4.3, which basic algorithm is the proposed method compared to, e.g. gradient-descent-ascent or its alternative version, or Adam, etc? What is the discrete version of the proposed method?


**Strength And Weaknesses:**

The strength of the paper is that the proposed methodology is very interesting and it provides a new way to deal with the circling issue in bilinear games. However, I am not fully convinced of the stability result, as well the convergence of the modified version of the controller for general GANs. Here are a few questions:
- What is the behavior of the modified version (in eq. 17) on Dirac-GANs? Is there still any convergence as in Theorem 2?
- When one talks about the stability of the training process, would it make more sense to check whether the losses of the discriminator and generator converge, compared to the use of FID or inception score? Somehow it is not clear why the training can be stable in terms of these scores.
- The discussion at the beginning of Section 3.2 regarding the optimal solution is too ideal. In practice, a normal GAN only has access to finite samples of p(x), so it is not clear why one can have p_G(x) = p(x), and D(x)=0 as an equilibrium. More seriously, the numerical results are based on the use of WGAN-GP, where the GP penalty does not make D(x)=0 (as it encourages that the gradient of D has a unit norm). Thus the idea of BRGAN seems not compatible with the GP, nevertheless wgan_gp_br seems to work best in Fig 2 and 3. I think more discussions are needed regarding these aspects to explain how wgan_gp_br works, and why it works well.

**Summary Of The Paper:**

This paper proposes a stochastic controller to stabilize the training process of GANs. It shows in a simple Dirac-GANs case, the proposed method converges theoretically to a unique optimal equilibrium. For general GANs, a modification of the controller is proposed and numerical results are given to show the improved stability of the training process in terms of common benchmarks for image generations.

**Summary Of The Review:**

I tend to stay with my current score, as I find that the current version could still be improved for a future submission.

---

> ### Author Response · Authors · 2022-11-17
> **Thank your for your supportive comments**
>
> We thank you for the insightful feedback. We address the detailed response as below. We hope you find your response satisfactory and raise your score accordingly.
>
> __Q1 Eq17 in Dirac-BrGANs__
> We use $\beta=2$ (as in equation 17) to conduct more experiment on Dirac-BrGANs. See general response A2.
>
> __Q2 Loss vs scores__
> We agree with the reviewer that the loss curve directly reflects the stability of GANs’ training process compared to inception and FID scores. However, in practice we found out as the score curves stabilize, so do the loss curves.
>
> _Q3 Equilibrium point of Discriminator__
> We understand that in practice gans do not have access to true data distribution and normally are unable to achieve theoretical Nash equilibrium at P_g = P_{data} and D(x) = 0. Our BMC’s goal is to push the discriminator to its Nash equilibrium at D(x) = 0. For gradient penalty, although the gp regularizer does not push D(x) to 0 to prevent the discriminator overrides the generator, the ultimate goal for the discriminator is to be indistinguishable, thus D(x) =0 for the nash equilibrium. We believe it works the best since our designed control makes D converge faster and gp helps alleviate the potential issue caused by stronger discriminator compared with weaker generator in DCGANs setting.
>
> __Miscellaneous questions at the end__
> Dot B_1 is the same as dB1. In control theory, we normally write the first derivatives of a variable using dot representation.
> The gradient penalty term is imposed on the discriminator. p_g in equation 13 is the distribution of the generator, which includes gp.
> Our proposed method in section 4.3 is on the objective function of the discriminator which can be applied to all optimizers.

---

### Official Review · Reviewer_y1pm · 2022-10-25

**Confidence:** 3
**Correctness:** 3
**Technical Novelty And Significance:** 3
**Empirical Novelty And Significance:** 2
**Recommendation:** 3

**Clarity, Quality, Novelty And Reproducibility:**

Writing could be improved, as space is not used wisely. For example, authors provide limited motivation and exposition on DiracGANs (Section 2.1), but spend space defining FID and Inception Score (Section 4.1), which are well known quantities. As a result, Section 2 required some outside reading to understand --- I had to refer to [2] for more information on DiracGAN. Even so, it is not clear to me why improving stability on the DiracGANs will improve general GAN stability.

Questions:

Tables 3 and 4 and Figures 4 and 5 — Why is WGAN and WGAN-BR performance so poor? The model does not appear to be converged to me.

A key contribution of this work is its connection to control theory. However, there is limited discussion on the relationship of this work to that of [1], and the fairness of the empirical comparisons is unclear. Therefore it is somewhat difficult to gauge novelty and quality. Can the authors elaborate on this?

Why is BrGAN only compared to CLC-GAN? Surely there are other appropriate regularizers / training schemes that aim to stabilize GAN training? If the authors choose to focus on only the control theory perspective, then the suitability of applying control theory to GAN training should be further justified.

[2] Mescheder, L., Geiger, A. and Nowozin, S., 2018, July. Which training methods for GANs do actually converge?. In International conference on machine learning (pp. 3481-3490). PMLR.

**Strength And Weaknesses:**

Strengths:
- Interesting perspective on GAN training.

Weaknesses:
- Theory is limited. Convergence results of the proposed method are restricted to the DiracGAN setting in a two-dimensional parameter space.
- Experiments could be more compelling. The authors claim that their method is an improvement over a previous work in a very similar direction that also applies control theory to GAN training -- WGAN-CLC [1]. However, the results reported in Table 3 deviate significantly from the results in [1]. Namely, [1] report much better FID scores than reported here. Can the authors comment on this discrepancy?
- Introduces three new hyperparameters $\rho_1$, $\rho_2$, and $\beta$. It is unclear how sensitive they are to perturbations, when applied to models trained on more realistic data (e.g. CIFAR-10, CelebA).

[1] Xu, K., Li, C., Zhu, J. and Zhang, B., 2020, November. Understanding and stabilizing GANs’ training dynamics using control theory. In International Conference on Machine Learning (pp. 10566-10575). PMLR.

**Summary Of The Paper:**

The authors leverage ideas from control theory to propose a regularizer, called the Brownian motion controller (BMC), which aims to stabilize GAN training. In practice, this amounts to an additive regularization term which is applied to the discriminator loss.

**Summary Of The Review:**

Overall, I found the direction of the work promising, but its execution wanting. It was unclear why the authors chose DiracGANs as a motivating example, and why the control theory perspective was the most compelling direction to alleviate the training instabilities. Moreover, experimental comparisons were weak, and limited to one competing regularizer (WGAN-CLC), applied to three GAN losses (WGAN, WGAN-GP, WGAN-CP).

---

> ### Author Response · Authors · 2022-11-17
> **Thank you for your supportive review**
>
> We appreciate your time and thoughtful feedback. We hope you find your response satisfactory and raise your score accordingly if possible.
>
> __Q1 Limited theory__
> We started with linear Dirac-GANs’ settings and gave theoretical deductions for BMC. Unlike Dirac-GANs, in normal GANs, we only know the Nash equilibrium for the discriminator but not for the generator. So our BMC only targets the discriminator and pushes it to equilibrium.
>
> __Q2 Experiment aligned with CLC__
> See general response A1
>
> __Q3 Robustness of Hyperparameter__
> See general response A2
>
>
> __Miscellaneous questions at the end__
> We pick Dirac-GAN as a motivating example because the issue of instability is firstly analyzed by Mescheder[1], where they propose Dirac-GAN to illustrate the unstable behaviors of various GANs. Our work focused on control theory’s perspective since stability analysis of differential systems is already well studied but few have linked it to the unstable nature of neural networks.
>
> We compared our method with CLC since they are the only work connecting control theory with GANs. Since our method can be considered as a noise based regularizer on the discriminator, we compared our work with WGAN-Clipping and WGAN-GP which also formatted as a regularizer on the discriminator. Currently we are working on combining our method with other GANs various and high resolution dataset and due to the time limit we will include our results on our final submission.
>
>
> [1] Lars Mescheder,  Andreas Geiger,  and Sebastian Nowozin.   Which training methods for gans do actually converge?    In International conference on machine learning, pp. 3481–3490. PMLR, 2018

---

### Official Review · Reviewer_sHwP · 2022-10-28

**Confidence:** 4
**Clarity, Quality, Novelty And Reproducibility:** The experimental results are less sat…
**Correctness:** 3
**Technical Novelty And Significance:** 2
**Empirical Novelty And Significance:** 2
**Recommendation:** 3

**Strength And Weaknesses:**

Strengths:
- The idea of taking GAN training as a dynamic system and applying Brownian Motion Controller on it sounds interesting.

Weaknesses:
- My biggest concern is about the experimental results. Only CIFAR-10 (32x32 resolution) and CelebA (64x64 resolution) are evaluated and the FID score is not state-of-the-art (actually there is a very big gap between the provided results and SOTA, eg. IS 5.42 on CIFAR-10 v.s. 9.18 in [1]). I truly understand the architecture may be different, but it would be better to see whether the proposed method can continually improve the performance on top of the best architecture. Given there is plenty of theory works studying GAN training stability issues, it is hard to trust this work can indeed help improve the GAN training process since this one considering its poor performance.
- Meanwhile, only two datasets are evaluated, and both of them are low-resolution. It would be better to show more results on LSUN, FFHQ, with their high-resolution version.
- The claim of "the training process of Dirac-BrGANs achieves exponential stability almost surely" sounds like an overclaim to me. To verify its correctness, I think some additional experiments are needed: e.g., using various hyperparameters (training G after training D every $N$ times), optimizers, batch size, and demonstrating that the proposed methods can help GAN training robust to different settings.
References:
[1] Differentiable Augmentation for Data-Efficient GAN Training

**Summary Of The Paper:**

This paper proposes a higher order Brownian Motion Controller (BMC) for BrGANs to stabilize GANs' training process. Starting with the prototypical case of Dirac-GANs, the authors design a BMC and propose Dirac-BrGANs that retrieve exactly the same but reachable optimal equilibrium regardless of GANs' framework. The authors also provide corresponding proof and experiments, although the experimental results show poor performance.


**Summary Of The Review:**

This work propose a new way to stabilize the training process of GAN, using Brownian Motion Controller. Although the motivation is interesting, the results are less satisfactory and the quality of the work is lower than the ICLR's bar.

---

> ### Author Response · Authors · 2022-11-17
> **Thank you for your valuable comments**
>
> Reviewer sHwP,
>
> We thank you for your valuable feedback and we address the detailed comments below.
>
>
> __Q1 GAP between provided result and SOTA, High resolution dataset__
> We aligned our experiment with the CLC setting (see general response A1) and due to the time limit, we will include our experiment with other SOTA GANs variations on high resolution dataset in our final submission.
>
> __Q2 Additional experiment for Dirac-BrGANs__
> See general response A2.

---

### Author Response · Authors · 2022-11-17
**General responses for all reviewers**

Dear Rewiers,

We appreciate all reviewers for their thoughtful comments. Here we address some common detailed responses. We look forward to any further feedbacks.

__A1. Aligned with CLC experiment__ [R1, R2, R4]
We redo our experiment on CLC’s setting and re-evaluate the performance of BrGANs and CLC using the same metric (Chiner for Inception Score [1], pytorch-FID for FID [2]), and the scores for CLC are similar to their original paper. The difference in our first draft was because we used a smaller number of generated images to test for those two scores.

_Table 1. Inception Score and FID score on Cifar 10 dataset (averaged over 3 runs)_

|          | IS| FID |
|----------|-----------------|-----|
| WGAN-CLC |     8.43 ± 0.13              |   21.08 ± 1.56     |
| WGAN-Br  |       8.67 ± 0.08               |              18.83 ± 0.72   |


__A2. Additional experiment for Dirac-BrGANs__[R1, R2, R3]
Distributions of training and generated samples are tractable in Dirac-GAN problem, so we can analyze stability and convergence in theory. Experiments should conform to our theoretical analysis because their settings are the same. We try batchsize 32, 64, 128, 256 and training G after training D every 1,2,3,5 time according to [R1]’s suggestion. Dirac-BrGANs converge within 2000 steps in all these settings.


Best,
Authors


[1] Huikai Wu, Shuai Zheng, Junge Zhang, and Kaiqi Huang. Gp-gan: Towards realistic high-resolution image blending. ACMMM, 2019
[2] Maximilian Seitzer. pytorch-fid: FID Score for PyTorch.https://github.com/mseitzer/pytorch-fid, August 2020. Version 0.2.1

---

### Decision · Program_Chairs · 2023-01-20

**Decision:**

Reject

**Justification For Why Not Higher Score:**

Reviews are clearly negative overall.

**Justification For Why Not Lower Score:**

N/A

**Metareview: Summary, Strengths And Weaknesses:**

This paper proposes to use a Brownian Motion Controller (BMC) to stabilize the GANs' training process. The stability of GANs is an important problem in the field so this is a relevant problem. On the plus side, the reviewers also find the idea to be interesting so I would encourage the authors to further pursue it and improve the execution of the paper. For now, the paper is still not at the level of an ICLR publication and I'm not able to recommend acceptance.

The most important comments from the reviewers seem to point to the following avenues for improvement:
- possibly strengthen the theory of the paper
- or strengthen the empirical evaluation
- better justify the choice of hyper-parameters, run experiments with multiple seeds, better describe the experimental setting in the text



**Summary Of Ac-Reviewer Meeting:**

N/A